# Large piezoelectric response in a Jahn-Teller distorted molecular metal halide

Sasa Wang[1,7], Asif Abdullah Khan[2,3,7], Sam Teale [1,7], Jian Xu[1,7], Darshan H. Parmar [1], Ruyan Zhao[4], Luke Grater [1], Peter Serles [5], Yu Zou [6], Tobin Filleter [5], Dwight S. Seferos[4], Dayan Ban [2,3] & Edward H. Sargent [1]

Piezoelectric materials convert between mechanical and electrical energy and are a basis for self-powered electronics. Current piezoelectrics exhibit either large charge ($d_{33}$) or voltage ($g_{33}$) coefficients but not both simultaneously, and yet the maximum energy density for energy harvesting is determined by the transduction coefficient: $d_{33}*g_{33}$. In prior piezoelectrics, an increase in polarization usually accompanies a dramatic rise in the dielectric constant, resulting in trade off between $d_{33}$ and $g_{33}$. This recognition led us to a design concept: increase polarization through Jahn-Teller lattice distortion and reduce the dielectric constant using a highly confined 0D molecular architecture. With this in mind, we sought to insert a quasi-spherical cation into a Jahn-Teller distorted lattice, increasing the mechanical response for a large piezoelectric coefficient. We implemented this concept by developing EDABCO-CuCl$_4$ (EDABCO = N-ethyl-1,4-diazoniabicyclo[2.2.2]octonium), a molecular piezoelectric with a $d_{33}$ of 165 pm/V and $g_{33}$ of ~2110 × 10$^{-3}$ V m N$^{-1}$, one that achieved thusly a combined transduction coefficient of 348 × 10$^{-12}$ m$^3$ J$^{-1}$. This enables piezoelectric energy harvesting in EDABCO-CuCl$_4$@PVDF (polyvinylidene fluoride) composite film with a peak power density of 43 μW/cm$^2$ (at 50 kPa), the highest value reported for mechanical energy harvesters based on heavy-metal-free molecular piezoelectric.

Piezoelectric energy harvesters have attracted attention for their mechanical-electrical conversion capability, light weight, low cost, and ready fabrication[1–3]. The electrical energy that a piezoelectric energy harvester generates depends on the product of the piezoelectric charge coefficient ($d_{33}$) and the piezoelectric voltage coefficient ($g_{33}$), the transduction coefficient ($d_{33}*g_{33}$)[4,5].

Oxide perovskites such as PZT (lead zirconium titanate) exhibit impressive $d_{33}$ values, the result of delocalized spontaneous polarization about a morphotropic phase boundary. However, this leads to a large increase in the dielectric constant, which limits their $g_{33}$ ($g_{33} = d_{33}/(\varepsilon_r \cdot \varepsilon_0)$)[6–8]. Although piezoelectric composites (high $d_{33}$ ceramics mixed with low dielectric constant polymers) have been used to address this problem, the $g_{33}$ of these composites remains to be further improved[9]. Molecular metal halide piezoelectrics, which include metal halide perovskites, are attractive materials for enhancing the transduction current of piezoelectric energy harvesters because they combine the benefits of organic and inorganic moieties at the molecular level[10,11]. In contrast to their rigid inorganic cousins,

[1]Department of Electrical and Computer Engineering, University of Toronto, 10 King's College Road, Toronto, Ontario M5S 3G4, Canada. [2]Waterloo Institute for Nanotechnology, University of Waterloo, 200 University Ave West, Waterloo, Ontario N2L 3G1, Canada. [3]Department of Electrical and Computer Engineering, University of Waterloo, 200 University Ave West, Waterloo, Ontario N2L 3G1, Canada. [4]Department of Chemistry, University of Toronto, 80 St. George Street, Toronto, Ontario M5S 3H6, Canada. [5]Department of Mechanical & Industrial Engineering, University of Toronto, 5 King's College Road, Toronto, Ontario M5S 3G8, Canada. [6]Department of Materials Science and Engineering, University of Toronto, 184 College Street, Toronto, Ontario M5S 3E4, Canada. [7]These authors contributed equally: Sasa Wang, Asif Abdullah Khan, Sam Teale, Jian Xu. ✉e-mail: dban@uwaterloo.ca; ted.sargent@utoronto.ca

molecular metal halides exhibit mechanical flexibility and low dielectric constants, providing an avenue for structural engineering united with strong piezoelectric response. However, the trade-off between $d_{33}$ and $g_{33}$ is still apparent in molecular metal halides.

Piezoelectricity arises in crystalline materials without inversion symmetry and is caused by polarization changes due to lattice distortion under stress. Ideally, a material will exhibit either strong polarization or large crystallographic distortion under stress. Reports have shown that plastic ferroelectric crystals with spherical molecular structures exhibit prominent piezoelectricity due to the mechanical softness of their lattices[12]. However, plastic crystals tend to crystallize in highly symmetric cubic crystal systems due to the disordered orientation of spherical molecules, making it challenging to obtain non-centrosymmetric structures[13].

We see that quasi-spherical theory, which explains how subtle modifications to spherical molecules can impart symmetry breaking into this materials system and cause significant polarization changes under external stress, provides a means to account for high $d_{33}$[14-16]. The Jahn-Teller distortion arising due to transition metal ions has been noted to induce low-symmetric geometric distortions by promoting degenerate electronic states, which in turn leads to intriguing properties including lattice distortion and high mechanical softness[17,18].

We reasoned that combining Jahn-Teller distortion within a framework of quasi-spherical cations could provide a non-centrosymmetric crystal structure that is more easily deformed under mechanical stress; thus, strong piezoelectric properties could be anticipated.

We employed quasi-spherical cation N-ethyl-1,4-diazoniabicyclo[2.2.2]octonium (EDABCO) as the A-site organic molecule, achieved by adding an ethyl group to break the mirror plane of the 1,4-diazabicyclo[2.2.2]octane (DABCO) molecule; splitting the $\sigma_h$ mirror plane results in a lower $C_{3v}$ symmetry, giving rise to molecular dipole moment (Fig. 1a).

With the goal of achieving a significant Jahn-Teller effect, we noted that the $3d^9$ $Cu^{2+}$ could induce pronounced Jahn-Teller distortion because of the unequal occupancy of degenerate orbitals[17]. Hence, we proposed that combining a quasi-spherical cation with Jahn-Teller active $Cu^{2+}$ ions should improve the piezoelectric response of the resultant molecular metal halide and increase the energy yield of a piezoelectric energy harvester.

## Results

### Designing molecular piezoelectrics

We prepared single crystals of EDABCO-CuCl$_4$ using a slow cooling method (Figure S1a) and determined the crystal structure using single-crystal X-ray diffraction (XRD). The phase purity of as-grown crystals was confirmed by powder XRD, which fit well with the simulated pattern (Figure S1b). Structural analysis reveals that EDABCO-CuCl$_4$ crystallizes in a non-centrosymmetric space group $P2_12_12_1$, among the 20 piezoelectric active point groups (Table S3). We observed second harmonic generation (SHG) (Figure S2) from EDABCO-CuCl$_4$, in contrast with its spherical analog, DABCO-CuCl$_4$, which crystallizes in the centrosymmetric crystal structure of $P2_1/c$[19]. These results agree with a picture in which the EDABCO cation contributes to symmetry breaking.

To gain insight into the role of Jahn-Teller distortion on symmetry breaking, we synthesized EDABCO-ZnCl$_4$, where the $3d^{10}$ $Zn^{2+}$ is Jahn-Teller inactive (Fig. 1b). Single-crystal structure analysis indicates that EDABCO-ZnCl$_4$ adopts a 0D structure composed of $[ZnCl_4]^{2-}$ tetrahedrons and EDABCO cations, similar to that of the EDABCO-CuCl$_4$ (Fig. 1c). However, EDABCO-ZnCl$_4$ belongs to the centrosymmetric space group $P2_1/c$[20]. We evaluated structural distortion using the geometry index $\tau_4 = \frac{360° - (\alpha + \beta)}{141°}$, where $\alpha$ and $\beta$ are the two largest Cl–Cu–Cl (or Cl–Zn–Cl) angles of the four-coordinate species[21,22]. Cl–Cu–Cl angles exhibit significant variation away from the ideal 109.5° (95.87° to 132.36°, Table S5) for a $\tau_4$ of 0.88 (where $\tau_4 = 1$ for a perfectly symmetric structure), while the Cl–Zn–Cl angles show only slight distortion (108.11° to 115.31°) for a $\tau_4$ of 0.94. This suggests that the lattice distortion originates, in part, from the Jahn-Teller effect of the incompletely filled $3d$ electronic configuration of the $Cu^{2+}$ ($3d^9$) ion.

### Investigation of the Jahn-Teller effect

We used density functional theory (DFT) calculations and steady-state optical absorption to elucidate the Jahn-Teller effect in EDABCO-CuCl$_4$. The band structure and Brillouin zone calculated

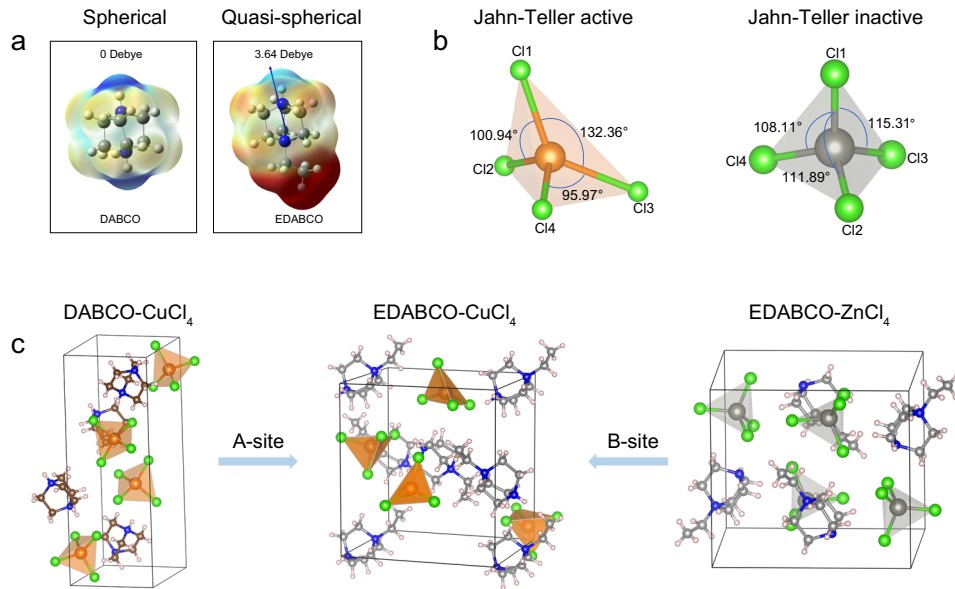

**Fig. 1 | Molecular design for piezoelectric materials. a** The modification of a spherical cation into a quasi-spherical cation lowers molecular symmetry and enables dipole moment. **b** The selected Cl–Cu–Cl and Cl–Zn–Cl bond angles of the $[CuCl_4]^{2-}$ and $[ZnCl_4]^{2-}$ tetrahedrons, representing those that deviate from the ideal 109.5°. The Jahn-Teller active $Cu^{2+}$ enhances the deformation of the inorganic geometry. **c** Comparison of the crystal structures of DABCO-CuCl$_4$, EDABCO-CuCl$_4$, and EDABCO-ZnCl$_4$. The combination of quasi-spherical cation and Jahn-Teller distortion results in the piezoelectric active structure.

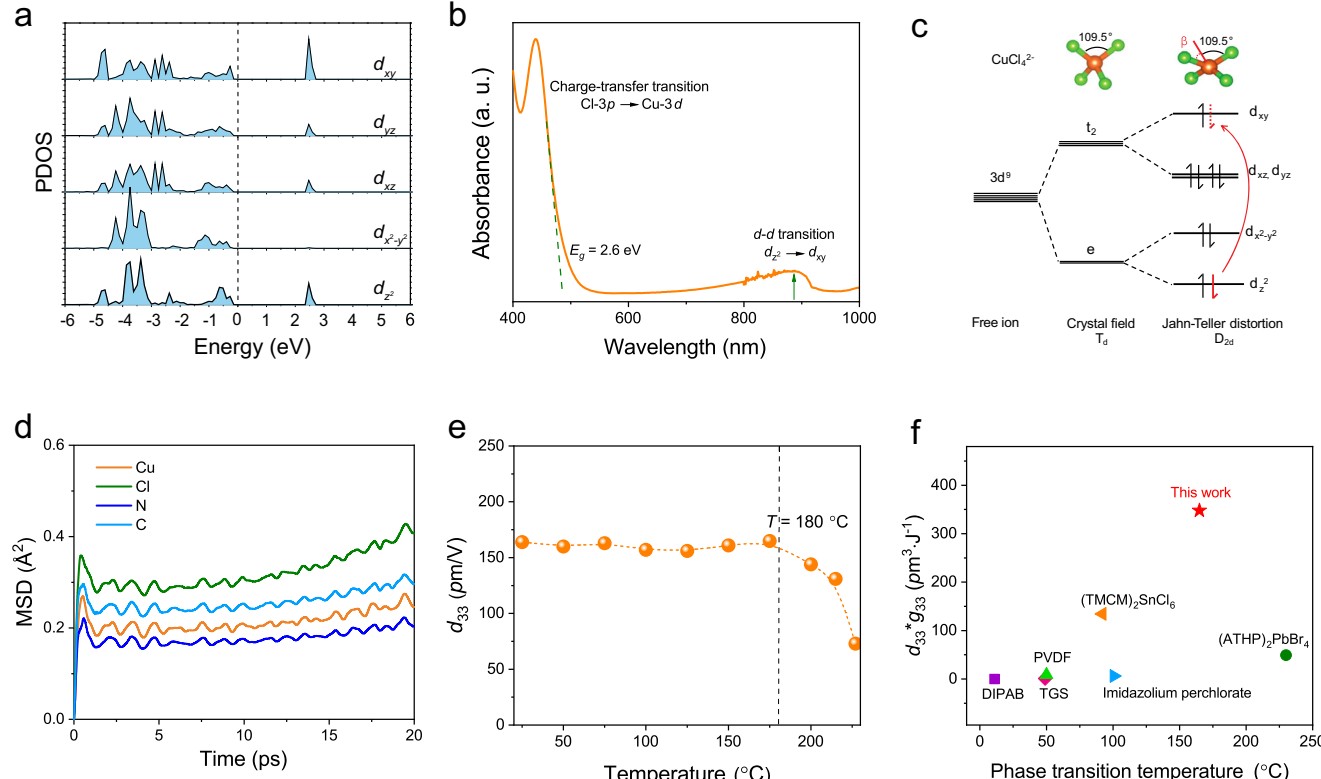

**Fig. 2 | Electronic structure and piezoelectric response of EDABCO-CuCl₄.**
**a** Projected density of states (PDOS) of Cu²⁺−3*d* orbitals in EDABCO-CuCl₄. **b** Optical absorption spectrum of EDABCO-CuCl₄ exhibits two bands, a visible band associated with charge transfer and an infra band ascribed to *d−d* transition. **c** Energy level diagram for the Cu²⁺−3*d* electronic configuration in a tetragonal (*D₂d*) crystal field showing the effect of Jahn-Teller distortion on the energy level splitting. **d** The averaged MSDs for Cu, Cl, N, and C atoms from AIMD simulations at 453 K for 20 ps. **e** Piezoelectric coefficient (*d₃₃*) of EDABCO-CuCl₄ single crystals as a function of temperature. **f** Comparison of solid-state phase transition temperature and $d_{33}*g_{33}$ with previously reported molecular piezoelectrics.

using HSE06 functionals are shown in Figure S3. From the projected density of states (PDOS) of the Cu²⁺−3*d* orbitals (Fig. 2a), we find that the triple-degenerate $t_2$ *d*-levels split into two double-degenerate states $d_{xz}$, $d_{yz}$ and a single state $d_{xy}$; the double-degenerate *e* *d*-levels split into a single state $d_{z^2}$ and $d_{x^2-y^2}$, which is attributed to the Jahn-Teller distortion. The optical absorption spectrum consists of two intense bands (Fig. 2b). The visible band at 2.6 eV is associated with a ligand-to-metal charge transfer consistent with the bandgap from DFT. The secondary, near-infrared peak corresponds to *d−d* electronic transitions within the tetrahedral crystal field of [CuCl₄]²⁻, where the local symmetry of Cu²⁺ is reduced from $T_d$ to $D_{2d}$ by Jahn-Teller distortion. The Cu²⁺−3*d* orbitals energy level splitting is illustrated in Fig. 2c.

We performed ab initio molecular dynamics (AIMD) simulations to evaluate the structural stability of EDABCO-CuCl₄ at 453 K (well above the automotive norm for piezoelectric stability, ~423 K)[23–25]. The averaged mean-squared displacements (MSDs) of AIMD simulations for 20 ps at 453 K show that Cu, Cl, N, and C atoms have constant small values with respect to their original equilibrium positions (Fig. 2d), which indicates that the structure of EDABCO-CuCl₄ in $P2_12_12_1$ symmetry is stable and no solid-state structural transformation is observed (Figure S4). We used TG-DSC analysis to characterize the thermal stability of EDABCO-CuCl₄ (Figure S5a). The DSC curve shows endothermic/exothermic peaks at 200–226 °C, due to the melting and the subsequent decomposition of EDABCO-CuCl₄. The decomposition process is seen in weight loss at 210 °C in the TG curve. These results are consistent with temperature-dependent XRD patterns (Figure S5b), where no phase transition occurs before 200 °C, suggesting the potential of EDABCO-CuCl₄ to operate in high temperatures.

## Assessing the piezoelectric response

We measured the piezoelectric coefficient ($d_{33}$) using the quasi-static method on a single crystal of EDABCO-CuCl₄[26]. For the $P2_12_12_1$ phase (point group 222), the piezoelectric coefficient matrix is as follows[11,27]:

$$\begin{pmatrix} 0 & 0 & 0 & d_{14} & 0 & 0 \\ 0 & 0 & 0 & 0 & d_{25} & 0 \\ 0 & 0 & 0 & 0 & 0 & d_{36} \end{pmatrix}$$

A longitudinal $d_{33}$ in the [111]-direction is possible, which is equal to $d_{14} + d_{25} + d_{36}$. We obtained a maximum $d_{33}$ of -165 pm/V along the [111] direction at room temperature (Figure S6), which shows no significant decrease until the temperature was increased to above 180 °C (Fig. 2e). This is significant as molecular metal halides tend to suffer from a low Curie temperature (typically <150 °C), so they cannot meet the operational requirements specified for automotive applications. We attribute the high piezoelectric coefficient of EDABCO-CuCl₄ to the increased mechanical flexibility, as manifested in Young's modulus measurements (Figure S7). The Young's modulus of EDABCO-CuCl₄ is measured to be 14.9 ± 0.7 GPa, which is significantly smaller than those of typical piezoelectric ceramics, indicating favorable mechanical flexibility, but higher than other non-piezoelectric metal halides, which allows it to be robust for repeated cycling performance.

EDABCO-CuCl₄ shows a relatively low $\varepsilon_r$ of 7–9 within the frequency range from 1 kHz to 1 MHz with no obvious anomaly in the temperature range of 20 to 200 °C (Figure S8). The low dielectric constant we attribute to the high dielectric confinement in the 0D structure[28]. Its $g_{33}$ is 2110 × 10⁻³ V m N⁻¹ at 1 kHz, ~60× higher than PZT-based piezoelectric ceramics (Table S1). Given its impressive $g_{33}$ and $d_{33}$ values, the energy density per unit volume ($d_{33}*g_{33}$) of

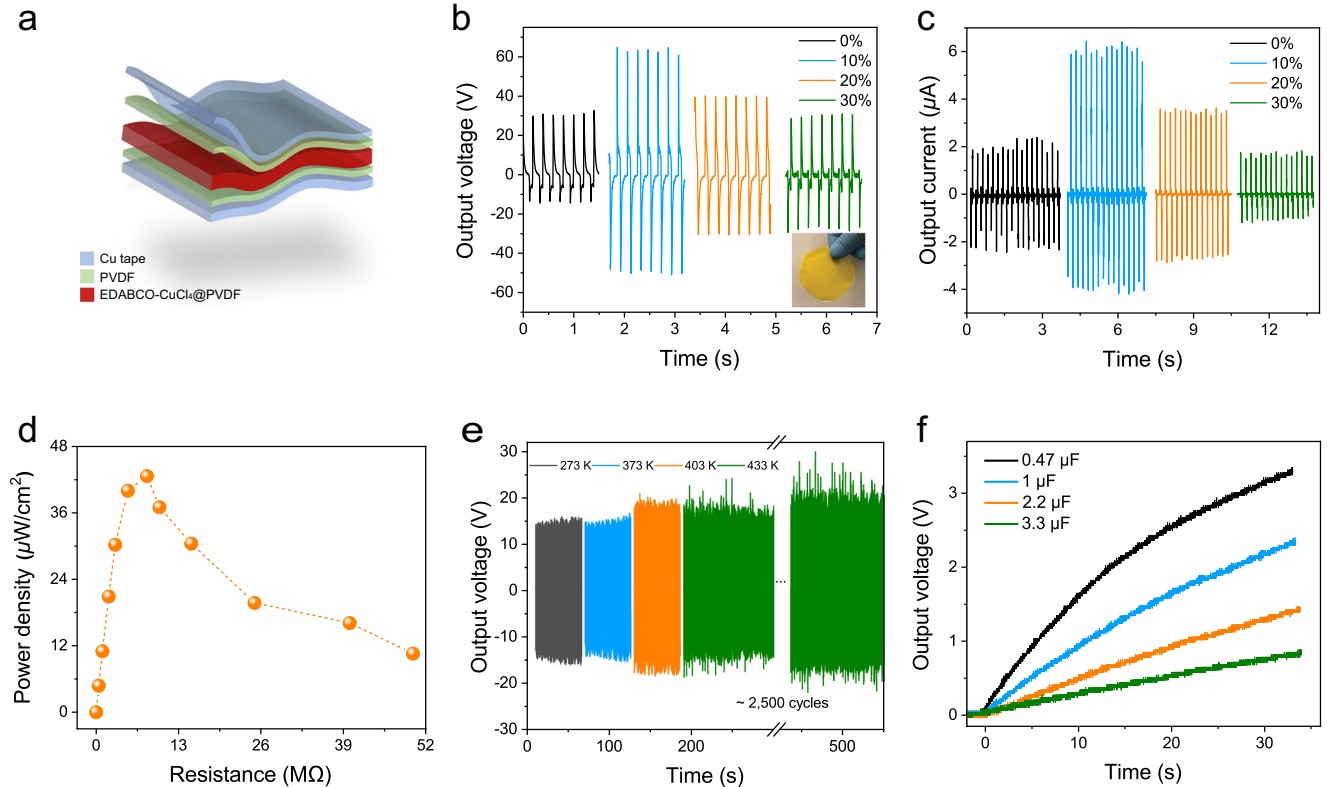

**Fig. 3 | Application of EDABCO-CuCl₄ in piezoelectric energy harvester. a** Schematic of a flexible piezoelectric energy harvester. **b** Output voltage and **c** current with different EDABCO-CuCl₄ concentrations at an applied force of 15 N and a frequency of 5 Hz. **d** The instantaneous power density on load resistances ranges from 0 to 50 MΩ. **e** The temperature-dependent output voltage of EDABCO-CuCl₄@PDMS composite films and reliability test for ~2500 cycles at 433 K. **f** Charging voltages of different capacitors using EDABCO-CuCl₄@PVDF piezoelectric generator.

EDABCO-CuCl₄ is evaluated to be $348 \times 10^{-12}\,m^3\,J^{-1}$, among the highest value for piezoelectric materials that do not contain regulated metals (Fig. 2f).

### Investigations of energy harvesting performance

The encouraging piezoelectric response of EDABCO-CuCl₄ motivated us to explore its energy-harvesting capability. Homogeneous polymer composites with 10, 20, and 30 wt% of EDABCO-CuCl₄ were prepared by dispersing the corresponding quantities in PVDF under vigorous mechanical stirring. PVDF is a stiff polymer (elastic modulus E = 0.03–17.1 GPa) with excellent mechanical strength and low viscosity for proper nanoparticle dispersion. Devices were fabricated by sandwiching the active composite films between two copper electrodes (Figs. 3a and S9).

The 10 wt% EDABCO-CuCl₄@PVDF devices generate the largest output, with a maximum output voltage ($V_{oc}$) of 63 V and a current density ($J_{sc}$) of 2.1 μA/cm² (15 N applied force, 5 Hz frequency) (Figs. 3b, c, and S10). Compared with a pure PVDF device, the incorporation of EDABCO-CuCl₄ nanoparticles increased the output voltage and current density by a factor of 2 and 3, respectively. The decrease in the output performance at higher EDABCO-CuCl₄ concentrations we attribute to particle aggregation, which leads to the randomization of dipoles (Figure S11)[29]. We performed switching-polarity tests to verify that the generated output signals originate from intrinsic piezoelectric properties (Figure S12a) rather than any electrostatic interactions. Fatigue experiments show no noticeable reduction in output current even after 35,000 mechanical pressing and releasing cycles, indicating high durability and potential for long-term operation (Figure S12b).

We measured the peak instantaneous output power density as a function of external load resistance from 0 to 50 MΩ (Fig. 3d). The maximum power density of the pure PVDF PG was measured to be 1.5 μW/cm² at a threshold load resistance of 2 MΩ (Figure S13). The 10 wt% piezoelectric generators (which contain only 5 mg of EDABCO-CuCl₄) produce a maximum power density of 43 μW/cm² at a threshold load resistance of 8 MΩ; the best reported for a heavy-metal-free molecular energy harvester (Table S2).

We sought to test the applicability of EDABCO-CuCl₄ for use in sensing and actuating in harsh environments. Polymer composites with 5 wt% EDABCO-CuCl₄ were prepared by dispersing the nanoparticles in polydimethylsiloxane (PDMS). PDMS is a heat-resistant flexible polymer for which thermal degradation occurs above 400 °C, in contrast to lower-melting-point PVDF (~160 °C). The average $V_{oc}$ (~20 V) was maintained following 2500 working cycles at 433 K, indicating its operability over a widened temperature range (Fig. 3e). When we increase the temperature from 273 to 433 K, we see an increase in the $V_{oc}$ (±5 V). This increase, as well as the higher pickup of ambient noise observed in Fig. 3e, warrants further mechanistic study, including a focus on the impact of temperature on both electrical conductivity and adhesive strength. The device and material exhibited no noticeable degradation following high-temperature operation (cross-sectional SEM and EDS mapping in Figure S14).

To demonstrate active energy harvesting, we stored the electrical energy from EDABCO-CuCl₄ in a series of capacitors (Fig. 3f). At a low frequency (5 Hz) the stored charges across the 0.47 μF capacitor reached an electronic node voltage level of ~3 V in less than half a minute, illustrating the potential for use as a power source for low-consumption micropower electronics.

### Discussion

The piezoelectric response of molecular metal halide EDABCO-CuCl₄ is enhanced by the combination of quasi-spherical theory and Jahn-Teller distortion. EDABCO-CuCl₄ has one of the largest energy density

potentials of any recorded material, and the resulting piezoelectric energy harvesters are highly stable. The 10 wt% EDABCO-CuCl$_4$@PVDF composites achieve output power densities superior to the best-reported molecular hybrid energy harvesters. This work contributes to progress in the design of heavy-metal-free molecular metal halides for high-performance piezoelectric energy harvesting.

## Methods

### Single crystal growth and structure characterization

1-Ethyl-1,4-diazabicyclo [2.2.2] octanium iodide (EDABCOI) was synthesized by the Menshutkin reaction. N,N′-diazabicyclo[2.2.2]octonium (DABCO) and ethyl iodide were reacted in acetone with a 1:1.2 mole ratio at room temperature for 24 h. The resulting precipitate was washed with a large amount of diethyl ether and collected by filtration.

Single crystals of EDABCO-CuCl$_4$ were grown by the slow-cooling method. 0.5 mmol of CuCl$_2$ and 0.5 mmol of EDABCOI were dissolved in 2.5 ml hydrochloric acid (38%). The mixture was placed on a hot plate and stirred at 80 °C for 30 min so that the saturated solution was totally dissolved. Then, the clear solution was put into an oven, the initial temperature was set to 80 °C, and the cooling rate was 1 °C/h. We obtained the orange bulk crystals by slowly cooling the solution to room temperature.

Single crystal X-ray diffraction data were collected at 290 K on a Bruker APEX-DUO diffractometer with graphite-monochromated Mo-Kα radiation (Burker Triumph, λ = 0.71073 Å). The APEX3 software was used for data reduction. The crystal structure was solved by direct method and then refined anisotropically by full-matrix least-squares on $F^2$ using the SHELXLTL software package. Hydrogen atoms of the EDABCO cations were generated in the geometrical mode. The detailed structural information is listed in Tables S3−S7. Deposited CCDC number: 2180969.

### XRD measurement

The powder XRD measurements were conducted on a Rigaku MiniFlex 600 diffractometer (Bragg-Brentano geometry) equipped with a monochromatized Cu Kα radiation source (λ = 1.5406 Å, 40 kV, and 15 mA). For temperature-dependent measurements, samples were heated using the TS 500 high-temperature attachment and data were collected at intervals of 20 °C.

### TG-DSC measurement

Thermogravimetric and differential scanning calorimetry (TG-DSC) measurements were performed on the ground powders of EDABCO-CuCl$_4$ using a NETZSCH STA 449F5 instrument. The compounds were sealed in aluminum hermetic pans between 25 and 800 °C at a heating rate of 10 °C•min$^{-1}$.

### SHG measurement

To measure second-harmonic generation (SHG), a regenerative amplifier made of ytterbium-doped gadolinium potassium tungstate was employed to produce a fundamental wave of 1030 nm at a frequency of 5 kHz. The resulting beam was directed through an optical parametric amplifier, specifically the Orpheus model from Light Conversion, to produce a wavelength of 1510 nm. The scattered light emanating from the sample was gathered using a pair of lenses and focused into a 400 μm multimode fiber (Thorlabs), which was then transmitted to a USB2000 + spectrometer produced by Ocean Optics for analysis.

### Dielectric permittivity and Young's modulus measurements

Bulk single crystals were used for dielectric permittivity and Young's modulus measurements. For dielectric permittivity measurement, the single crystal was sandwiched between two ITO substrates that served as top and bottom electrodes. The dielectric permittivity ($\varepsilon_r$) is measured on a potentiostat electrochemical workstation (AUT50690, PGSTAT204, The Netherlands) with a frequency ranging from 1 kHz to

1 MHz with an AC voltage of 0.5 V. Nanoindentation was performed on an iMicro Nanoindenter (KLA Corp., California, USA) using a diamond Berkovich tip to a depth of 800 nm in continuous stiffness mode. Young's modulus was extracted using the Oliver & Pharr method.

### Piezoelectric measurement

The macroscopic piezoelectric coefficient ($d_{33}$) was measured by a commercial piezometer (Polyk, PKD3-2000-F10N) using "Berlincourt" method (also called "quasi-static" method). The crystal was placed in between two flat metal plates, which clamped the sample and applied a small oscillating force (0.25 N, 110 Hz) along the normal direction while the piezoelectric charge was measured. For temperature-dependent measurements, the crystals were heated using a miniature temperature chamber, and an omega temperature controller controlled the temperature.

### Preparation of EDABCO-CuCl$_4$@PVDF composite film-based piezoelectric generator (PG)

The Young's modulus disparity between the host matrix and the NPs should be minimized in order to maximize the efficiency of stress transfer to the NPs. PVDF has Young's modulus value (0.03–17.1 GPa) that is much closer to that of EDABCO-CuCl$_4$ (14.7 ± 0.7 GPa) compared to the values in other polymers such as polyvinyl acetate (PVA), polycaprolactone (PCL), and thermoplastic polyurethane (TPU). We selected PVDF to reinforce the EDABCO-CuCl$_4$ NPs with this in mind.

To synthesize the composite film, PVDF powders (polyvinylidene fluoride ≥99.9%, Sigma-Aldrich) were dissolved in DMF (N, N-dimethylformamide ≥99%, Sigma-Aldrich), followed by stirring at room temperature (25 °C) for 24 h. The final concentration of the PVDF solution is kept at 10 wt% and the phase of PVDF is confirmed as beta-phase (Figure S15). Fine ground nanoparticles (NPs) of EDABCO-CuCl$_4$ with an average size of between 50–150 nm (Figure S16) were used to prepare the EDABCO-CuCl$_4$@PVDF composite solution. To optimize the concentration, 10, 20, and 30 wt% composite solutions were prepared by rigorously stirring the solution for 24 h, followed by ultrasonic stirring for 30 min. The as-prepared solutions were spin-coated on a clean glass substrate (cleaned with O$_2$ plasma treatment) for 20 s at 160 rpm. The spin-coated solution was stored for 30 min for degassing purposes. Then the solution was cured by a slow curing process at 75 °C on a flat hot plate. Finally, the film was annealed at 110 °C for 10 min and peeled off from the glass substrate. To explore the influence of poling on the output performance, we measured the output voltage of the EDABCO-CuCl$_4$@PVDF film before and after applying an electrical poling field of 50 V·μm$^{-1}$ for 2–3 h, and the results showed that poling did not affect the output performance (Figure S17).

To fabricate a PG (piezoelectric generator), four units of composite film were used. Two PVDF thin films were prepared by spin-coating 10 wt% PVDF solution at 400 rpm for 20 s. The composite films were sandwiched between the PVDF thin films by pressing them with a heat press machine at 160 °C for 2 min. Two copper electrodes were attached to the thermal laminating pouches (4 mills thick), and then the device was encapsulated between the electrodes by pressing them in a thermal laminator. Copper tape was attached to one of the surfaces of the thermal laminating pouches to minimize static charge buildup during the periodic application of force by a metallic hammer. A finite element simulation was performed to distinguish the projected piezoelectric potential from the PVDF, and EDABCO-CuCl$_4$@PVDF composites (Figure S18).

### Characterization of EDABCO-CuCl$_4$@PVDF composite film-based PG

The mechanical energy harvesting capability was tested by mounting the PG on an L-shaped steel stopper and applying a periodic compressing force of -15 N at 5 Hz from a custom-made linear motor. The output voltage was measured by a digital oscilloscope (Tektronix 2004

C) and a 100 X cable. To measure the short-circuit output current, a low-noise current preamplifier (SR 570, Stanford Research Systems Inc.) was used in this experiment. The output of the PG was rectified by a full-wave bridge rectifier circuit (Digi-Key Electronics, B80C800G-E4/51). The rectified output was directly fed to different commercial capacitors to store the charges. The output voltage of the capacitors was measured by the 100 X cable and recorded by the oscilloscope. The reliability testing of the PG was performed using an electro-dynamic shaker (ET-126-1). A controller unit (VR 9500) generates different control signals which are amplified by a power amplifier (Lab works Inc. pa 138) to regulate the vibration of the shaker. The controller unit was operated via a workstation interface (Vibration View 9). The device was attached on top of a hammer unit of the shaker with double-sided Kapton tape, and a metal block (steel) was placed on top of the device. The outer side of the metal block was bounded to a Kapton top which only allowed it to move vertically. The dielectric properties of the film were measured by using a Keithley-4200 semiconductor parameter analyzer. The morphology of the composite film was analyzed by using a field emission scanning electron microscope (Hitachi SU5000 SEM) capable of performing energy-dispersive X-ray spectroscopy (EDS).

### Synthesis of EDABCO-CuCl$_4$@PDMS-based PG
To fabricate the high-temperature stable EDABCO-CuCl$_4$-based PG, the host polymer matrix had to redesign with PDMS (Polydimethylsiloxane, Sylgard 184 Silicone Elastomer, Dow Corning), as it can operate from a temperature ranging from −45 °C to 200 °C (in contrast to a low melting point of -160 °C of PVDF). The piezoelectric composite solution was prepared by mixing the fine-ground EDABCO-CuCl$_4$ NPs in PDMS for 24 h with a mass ratio of 5 wt%. The mass ratio was kept to a lower level (5 wt%) for this case to prevent the clustering of the nanoparticles, and to preserve the flexibility of the PDMS.

Two 128 μm thick flexible Kapton films were chosen as the substrate due to their excellent thermal stability (−75 °C to 260 °C). The Kapton films were coated with Ag (silver) ink as the electrodes by the screen-printing method and were cured at 100 °C.

Before drop-casting the composite solution on the Ag electrodes, a curing agent was added to the mixture with a weight ratio of 1:10 with the PDMS base. Followed by further stirring and ultrasonication for 1 h, the solution was drop cast on the Ag-coated Kapton substrates and was degassed for 1 h. The solution was partially cured at 100 °C for ~30 min on a hot plate. Then the partially cured composite films were gently pressed together by the adhesion of PDMS. Finally, the device was completely cured in a vacuum oven at 90 °C overnight.

### Characterization of EDABCO-CuCl$_4$@PDMS-based PG
The cross-sectional SEM image and the EDS mapping illustrate the uniform adhesion between the different layers of the PG, thus alleviating the triboelectric signals from the device. Two electrical connections were brought out for the measurements by single-core insulated Al (Aluminum) cables. The electrical poling was performed at an electric field of 50–120 V μm$^{-1}$ for 2–3 h.

To investigate the thermal stability of the PG, it was directly mounted on an electric heater which was used as the stopper. A linear motor was used to generate a pushing force of ~15 N at 5 Hz on the PG. To eliminate the error between the set temperature and the actual temperature on the device, an infrared thermometer temperature gun was used to measure the PG temperature simultaneously. The generated output voltage from the PG was continuously monitored by a digital oscilloscope (Tektronix 2004 C).

### DFT simulation
The Vienna Ab initio Simulation Package (VASP) was employed to perform first-principles calculations based on density functional theory (DFT)[30]. The Perdew–Burke–Ernzerhof functional (PBE)[31] was used for the exchange-correlation functional, along with the screened Heyd–Scuseria–Ernzerhof (HSE) hybrid functional[32,33]. The Hartree-Fock term in HSE calculations was set to a mixing parameter (α) 0.25. To account for van der Waals (vdW) correction, we implemented the DFT-D3 method[34]. The plane-wave cut-off energy utilized was 400 eV, while the energy and force convergence criteria were set at 10$^{-5}$ eV and 0.02 eV·Å$^{-1}$, respectively.

To calculate the ligand's electrostatic potential and dipole moments, the Gaussian 09 package was utilized with DFT-D3 at the B3LYP/def2TZVP level. The resulting electrostatic potential plot was then visualized using GaussianView.

To simulate Ab initio molecular dynamic (AIMD), the CP2K package[35] was utilized in the constant-volume and constant-temperature (NVT) ensemble. The temperature was controlled with a Nosé–Hoover thermostat[35] at 453 K, with a time step of 1.0 fs. The PBE-D3 functional was employed in combination with double-zeta basis sets (DZVP-MOLOPT)[36] and Goedecker–Teter–Hutter (GTH) pseudopotentials[37], and the cut-off was set to 560 Ry. GROMACS was used to analyze the averaged mean-squared displacements (MSDs) from AIMD simulations[38].

## Data availability
The data supporting this study are available within the paper and its Supplementary Information. Additional information is available from the corresponding author upon reasonable request. Source data are provided with this paper.

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

## Acknowledgements
This work was supported by Sony (512312) and the Natural Sciences and Engineering Research Council (506511). The authors thank the Giga-to-Nanoelectronics Centre, University of Waterloo for the experiment facilities.

## Author contributions
E.H.S. and D.B. conceived and supervised the project. S.W. designed the experiment, synthesized the EDABCO, grew the crystals, and performed the XRD, SHG, and dielectric measurements. A.K. prepared the energy-harvesting thin film and performed the SEM and mechanical energy-harvesting measurements. S.W. and S.T. performed the piezoelectric measurements. J.X. performed the DFT calculations. R.Z. performed the TG-DSC measurement. P.S. performed the nanoindentation measurement. D.P. and L.G. assisted in data analysis. Y.Z., T.F., and D.S. supervised their team members involved in the project and provided feedback for the manuscript. All authors contributed to the discussion and interpretation of the results. S.W., A.K., S.T., and J.X. wrote the manuscript with comments from all authors.

## Competing interests
The authors declare no competing interests.
