## [Peer Review File · Nature Communications]

Large piezoelectric response in a Jahn-Teller distorted molecular metal halideREVIEWER COMMENTS

Reviewer #1 (Remarks to the Author):

The focus of this work is to explore the piezoelectric energy harvesting in EDABCO-CuCl₄ (EDABCO = N-ethyl-1,4-diazoniabicyclo[2.2.2]octonium). A large d_{33} of 165 pm/V and g_{33} of $2650 \times 10^{-3} \text{ V m N}^{-1}$ is achieved. The experimental analyses have been done carefully. It is a topic of interest to the researcher in the molecular ferroelectric areas. However, some of the results are unclear, it is recommended for publication on Nature Communications after addressing the following points.

1. Please explain why the results for Figure S11 and Figure 3b-3c are inconsistent. Specifically, output voltage peaks of about 40 V and -30 V are reported in Fig. 3b for 20 wt.% EDABCO-CuCl₄@PVDF, which is significantly different from the obtained output voltage peaks of about 15 V and -35 V are reported in Fig. 3b for the same composition. Same question is for the output currents in Figure S11b and Figure 3c.
2. For Figure S12, please provide the EDS image for Cu element. Please also explain why the EDS mapping results for Cl and Ag elements show a strongly overlapped line.
3. Please provided SEM images for the "fine ground EDABCO-CuCl₄ nanoparticles".
4. The method for charging voltages of different capacitors (Fig. 3f) is missing in the experimental section.
5. Line 80, "DABCO-CuCl₄" should be "EDABCO-CuCl₄".
6. It is suggested to add a temperature dependence of dielectric constant measurement results for EDABCO-CuCl₄ single crystal to show its dielectric transition temperature.

Reviewer #2 (Remarks to the Author):

The manuscript by Sargent, Ban and co-workers reports the synthesis of a new organic-inorganic hybrid EDABCO-CuCl₄ with a high piezoelectric coefficient and its mechanical energy harvesting studies. The noncentrosymmetric origin of this material arises from the quasi-spherical nature of the EDABCO cation, which however has insufficient asphericity to endow the material with ferroelectric properties. The presence of Jahn-Teller effects in the compound was probed by DFT calculations. The high d_{33} and low permittivity in this material gave rise to significantly high g_{33} and high volume energy density; yet, these are not the highest (see Nature Communications, 2022, 13, 5607 and references therein). The authors also report high mechanical energy harvesting output voltage and power density values based on its PVDF composite. Despite all these properties, this reviewer finds a number of flaws in this manuscript, particularly due to use of PVDF in the composite material. No doubt this molecule shows some good energy harvesting outputs but to the reviewer's knowledge the reported device outputs are a combined effect of both PVDF and the molecule. From Figure 3b, one gets the impression that the presence of hybrid salt enhances the properties of PVDF and makes it a better piezoelectric material to be used under ambient conditions. At this level of studies, this molecule does not possess any excellent novelty to be published in Nature Comm, owing to the fact that the use of Jahn-Teller anions such as CuX₄ and modified DABCO are well explored in hybrid ferro- and piezoelectric materials and that the reported high mechanical energy harvesting is the result of the combination of both hybrid salt and the PVDF polymer. I suggest the authors pay attention to the following points before it can be resubmitted to this journal or elsewhere.

Comments and questions

1. Why did the authors choose PVDF polymer for energy harvesting applications? Is there any particular reason for choosing PVDF? As PVDF is one of the most established piezoelectric polymers, using PVDF to make an energy harvester is only justified when the authors can clearly distinguish between the contributions of only PVDF and their own material.
2. Did the authors look into the phase of PVDF in their composites? As PVDF is piezoelectric for one of its forms, the authors should specify the actual phase of PVDF in their composites.
3. Additionally, the authors should highlight the increase in the output performance of PVDF after dispersing their crystallites in PVDF. Because from Figure 3b it is evident that PVDF itself is giving a voltage of 25 V. So, to the reviewer's knowledge, the actual output of the compound can be stated as only 38 V.
4. Even the power density has almost 40 % contribution from PVDF itself. So it is ambiguous to say that this molecule gives an extraordinary power density of $43 \mu\text{W}/\text{cm}^2$. Since there are several molecular nanogenerator devices reported, both with PVDF and other commercial amorphous polymers like poly dimethyl siloxane (PDMS) with some fair power density values, The authors should discuss all these parameters broadly to claim their highest power density.
5. Again. The statement "The 10 wt. % piezoelectric generators (which contain only 5 mg of EDABCO-CuCl₄) produce a maximum power density of $43 \mu\text{W}/\text{cm}^2$ at a threshold load resistance of 8 M Ω ; the best reported for a molecular perovskite energy harvester" by the authors is not justifying the output of the material solely, as PVDF has its own contributions.
6. How did the authors minimize the static charge developments on the surface of the device during the piezoelectric measurements?
7. The authors state that "With increasing temperature (273 K to 433 K) the VOC of the piezoelectric generators does not decrease, nor does it alter after ~2,500 working cycles, indicating its operability over a large temperature range (Figure 3e)". For the temperature-dependent Piezoelectric measurements, the authors chose PDMS, why? The authors could have chosen PDMS itself for all their nanogenerator measurements, the use of two different polymers (PVDF and PDMS) should be explained properly.
8. The space group- P212121 only allows for shear piezoelectricity (d₁₄/d₂₅/d₃₆). So, what could be the reason for this high piezoelectric coefficient value i.e 165 pm/V?
9. Jahn-Teller distortion is very well known in many of the organic-inorganic hybrid perovskites, so, Can the authors specify the novelty in this material?
10. What is the melting point of the compound?
11. It is mentioned that there is no phase transition in this molecule, but the variable temperature PXRD data clearly shows that some changes are happening in the molecule before 200 OC. The authors should describe the changes in the PXRD pattern, where new peaks are appearing after a certain temperature (117 to 121 line)
12. The authors mentioned that "With increasing temperature (273 K to 433 K) the VOC of the piezoelectric generators does not decrease, nor does it alter after ~2,500 working cycles, indicating its operability over a large temperature range (Figure 3e)." Figure 3e shows that the cyclic stability measurements at 433K result in a change in the voltage (~20 V to 30 V), what could be the reason for this? And why the authors are still claiming that there is no change associated.
13. Please provide the method employed for the construction of the surface and provided dipole moment values in Figure 1a.

Reviewer #3 (Remarks to the Author):

Piezoelectrics with both large d_{33} and large g_{33} are highly desirable for piezoelectric energy harvesting. In this manuscript, the authors reported a non-ferroelectric organic-inorganic hybrid Cu(II) halide piezoelectric EDABCO-CuCl₄, which shows a d_{33} and g_{33} up to 165 pm/V and $\sim 2650 \times 10^{-3} \text{ V m N}^{-1}$, respectively. The EDABCO-CuCl₄-based piezoelectric energy harvester exhibits excellent performance with a high peak power density of 43 $\mu\text{W}/\text{cm}^2$. It is known that large d_{33} piezoelectrics are generally based on ferroelectrics. The discovery of a non-ferroelectric with such a high d_{33} of 165 pm/V is interesting. I think this work gives new insights into the piezoelectric field and provides a good strategy for the exploration of high-performance piezoelectrics. However, before publication in Nature Communications, the following issues should be addressed.

1. Although there are many kinds of perovskite structures ranging from 0D-3D, they are derived from the typical perovskite CaTiO₃ and basically have octahedral configurations in the structure. Therefore, it is hard to regard EDABCO-CuCl₄ with [CuCl₄]²⁻ tetrahedrons as perovskites.
2. The authors also mentioned the enhanced piezoelectric response in the title. But the DABCO-CuCl₄ and EDABCO-ZnCl₄ used for comparison have no piezoelectric response.
3. In the abstract, the authors stated that the quasi-spherical cation into a Jahn-Teller distorted perovskite lattice increases the mechanical strain for a large piezoelectric response, while the strain data was not provided.
4. The strategy by modification of a spherical cation into a quasi-spherical cation to lower molecular symmetry to regulate the crystal symmetry has recently been proposed as the quasi-spherical theory (J. Am. Chem. Soc. 2020, 142, 15205-15218). This should be mentioned in the manuscript.
5. The authors obtained the d_{33} along the [111] direction. Please provide the corresponding data to verify the [111] direction.
6. Generally, the calculation of g_{33} is based on the ϵ_r at the frequency of 1 kHz.
7. The authors prepared the EDABCO-CuCl₄@PVDF composite film-based piezoelectric generator. However, PVDF is also piezoelectric. How to rule out the influence of PVDF.
8. In the experimental section, for the preparation of EDABCO-CuCl₄ single crystals, 1-Ethyl-1,4-diazabicyclo [2.2.2] octanium iodide was used. The authors are suggested to check whether the iodide ion was doped into the single crystals. It is better to use the 1-Ethyl-1,4-diazabicyclo [2.2.2] octanium chloride to prepare EDABCO-CuCl₄.
9. The authors used high-voltage electrical poling to align the dipoles in the EDABCO-CuCl₄@PVDF film. But EDABCO-CuCl₄ is a non-ferroelectric.
10. In Table S1, some molecular piezoelectrics with large d_{33} and large g_{33} were not listed, such as the C₆H₅N(CH₃)₃CdBr₂Cl_{0.75}I_{0.25} (Nat. Commun. 2022, 13, 5607).

Actions in response to reviewer comments

Reviewer #1

Reviewer comment: The focus of this work is to explore the piezoelectric energy harvesting in EDABCO-CuCl₄ (EDABCO = N-ethyl-1,4- diazoniabicyclo[2.2.2]octonium). A large d_{33} of 165 pm/V and g_{33} of 2650×10^{-3} V m N⁻¹ is achieved. The experimental analyses have been done carefully. It is a topic of interest to the researcher in the molecular ferroelectric areas. However, some of the results are unclear, it is recommended for publication on Nature Communications after addressing the following points.

1. Please explain why the results for Figure S11 and Figure 3b-3c are inconsistent. Specifically, output voltage peaks of about 40 V and -30 V are reported in Fig. 3b for 20 wt.% EDABCO-CuCl₄@PVDF, which is significantly different from the obtained output voltage peaks of about 15 V and -35 V are reported in Fig. 3b for the same composition. Same question is for the output currents in Figure S11b and Figure 3c.

Response: We thank the reviewer for pointing out this issue. It has led us to correct the manuscript in these ways:

- In Figure S11a, the output voltage during a polarity switch was for a pure PVDF PG, but in the caption unfortunately we wrote it as 20 wt.% EDABCO-CuCl₄@PVDF PG. We have corrected this figure (now Figure S12a), and it now shows the output current before and after a polarity switch for the 10 wt. % EDABCO-CuCl₄@PVDF PG.
- We now better explain that the difference between Figure S11b and Figure 3c is due to a difference of the frequency and input force used in the different measurement setups. We used a linear motor and a fixed hammer for different weight ratio measurements, but the measurement frequency of this setup is limited to a maximum of 8 Hz. To mimic the vibration for thousands of cycles, we used an electrodynamic shaker for the device reliability test, which can generate a vibration of up to several kHz.

We have added the following description to the revised manuscript (page 15):

"The reliability testing of the PG was performed using an electrodynamic shaker (ET-126-1). A controller unit (VR 9500) generates different control signals which are amplified by a power amplifier (Lab works Inc. pa 138) to regulate the vibration of the shaker. The controller unit was operated via a workstation interface (Vibration View 9). The device was attached on top of a hammer unit of the shaker with double-sided Kapton tape, and a metal block (steel) was placed on top of the device. The outer side of the metal block was bounded to a Kapton top which only allowed it to move vertically."

Supplementary Figure S12. (a) Polarity switching of the output current of the 10 wt.% EDABCO-CuCl₄@PVDF PG. (b) The device reliability test for 35,000 cycles of the 10 wt.% EDABCO-CuCl₄@PVDF PG at 30 Hz with a 136-gram weight on top of the device.

2. For Figure S12, please provide the EDS image for Cu element. Please also explain why the EDS mapping results for Cl and Ag elements show a strongly overlapped line.

Response: When we fabricated the device, we used a composite solution of the EDABCO-CuCl₄@PDMS as the adhesive between the Ag electrode and the main composite film. Therefore, the surface of the Ag electrode is covered with the EDABCO-CuCl₄@PDMS. Furthermore, the characteristics of the X-ray energy of Ag and Cl are close (Ag L-alpha ~ 2.98 keV and Cl k-alpha 2.62 KeV). Thus, we have an overlap between Ag and Cl in the EDS mapping image.

We have now increased the processing time and channels (512 vs 1024) in the new EDS measurement to capture high-resolution images and distinguish the Ag and Cl element. We can see that the overlap has been reduced, and thus Ag and Cl can be distinguished clearly.

We have added the new SEM image and EDS mapping results in Figure S14 (updated figure number) to the revised manuscript.

Supplementary Figure S14. (a) The cross-sectional SEM and (b) EDS mapping of EDABCO-CuCl₄@PDMS piezoelectric generator after operating at 433 K for ~2,500 cycles.

3. Please provided SEM images for the “fine ground EDABCO-CuCl₄ nanoparticles”.

Response: We carried out the suggested measurements, and the SEM image is added to the revised manuscript and supporting information (see Figure S16 for detail).

We have added the following sentence to the revised manuscript (page 14)

“Fine ground nanoparticles (NPs) of the EDABCO-CuCl₄ single crystal with an average size of between 50-150 nm (Figure S16) was used to prepare the EDABCO-CuCl₄@PVDF composite solution.”

Supplementary Figure S16. SEM image of the finely ground EDABCO-CuCl₄ nanoparticles.

4. The method for charging voltages of different capacitors (Fig. 3f) is missing in the experimental section.

Response: We have now added the following sentence in the experimental section (page 15):

“The output of the PG was rectified by a full-wave bridge rectifier circuit (Digi-Key Electronics, B80C800G-E4/51). The rectified output was directly fed to different commercial capacitors to store the charges. The output voltage of the capacitors was measured by the 100 X cable and recorded by the oscilloscope.”

5. Line 80, “DABCO-CuCl₄” should be “EDABCO-CuCl₄”.

Response: We use the DABCO-CuCl₄ as a control to demonstrate that EDABCO plays an essential role in generating symmetry breaking.

We have now rewritten the sentence in the revised manuscript (page 5):

“This contrasts with its spherical analogue, DABCO-CuCl₄, which crystallizes in the centrosymmetric crystal structure of *P2₁/c*. These results indicate that the EDABCO cation plays an important role in the generation of symmetry breaking.”

6. It is suggested to add a temperature dependence of dielectric constant measurement results for EDABCO-CuCl₄ single crystal to show its dielectric transition temperature.

Response: We now provide the temperature-dependent dielectric constant of EDABCO-CuCl₄, shown below and in the revised manuscript Figure S8b. It is shown that no obvious anomaly of the dielectric constant is observed in the temperature range of 20 to 200 °C, indicating that no solid-state phase transition occurs.

Supplementary Figure S8. Temperature-dependent data of the dielectric constant of EDABCO-CuCl₄ at different frequencies.

Reviewer #2 (Remarks to the Author):

The manuscript by Sargent, Ban and co-workers reports the synthesis of a new organic-inorganic hybrid EDABCO-CuCl₄ with a high piezoelectric coefficient and its mechanical energy harvesting studies. The noncentrosymmetric origin of this material arises from the quasi-spherical nature of the EDABCO cation, which however has insufficient asphericity to endow the material with ferroelectric properties. The presence of Jahn-Teller effects in the compound was probed by DFT calculations. The high d_{33} and low permittivity in this material gave rise to significantly high g_{33} and high volume energy density; yet, these are not the highest (see Nature Communications, 2022, 13, 5607 and references therein). The authors also report high mechanical energy harvesting output voltage and power density values based on its PVDF composite. Despite all these properties, this reviewer finds a number of flaws in this manuscript, particularly due to use of PVDF in the composite material. No doubt this molecule shows some good energy harvesting outputs but to the reviewer's knowledge the reported device outputs are a combined effect of both PVDF and the molecule. From Figure 3b, one gets the impression that the presence of hybrid salt enhances the properties of PVDF and makes it a better piezoelectric material to be used under ambient conditions. At this level of studies, this molecule does not possess any excellent novelty to be published in Nature Comm, owing to the fact that the use of Jahn-Teller anions such as CuX₄ and modified DABCO are well explored in hybrid ferro- and piezoelectric materials and that the reported high mechanical energy harvesting is the result of the combination of both hybrid salt and the PVDF polymer. I suggest the authors pay attention to the following points before it can be resubmitted to this journal or elsewhere.

Comments and questions

1. Why did the authors choose PVDF polymer for energy harvesting applications? Is there any particular reason for choosing PVDF? As PVDF is one of the most established piezoelectric polymers, using PVDF to make an energy harvester is only justified when the authors can clearly distinguish between the contributions of only PVDF and their own material.

Response: We now better explain our selection of PVDF for its lower melting point (~ 160 °C) and higher elastic modulus (0.03-17.1 GPa) compared to typical thermoplastic polymers. Because of this material's low melting point, we were able to perform layer-by-layer assembly of thin films to fabricate PG using thermal pressing techniques. Most thermoplastic polymers have a high melting point beyond 200 °C, and if our process temperature is higher than 180 °C will degrade the d_{33} of EDABCO-CuCl₄. In addition, PVDF has low viscosity in organic solvents, which is favourable for nanoparticle dispersion. Indeed, PVDF is a well-known piezoelectric polymer, and it has contributed to mechanical energy harvesting (*Nat. Commun.*, **2020**, *11*, 1030; *Nano Energy*, **2020**, *78*, 105251; *Adv. Mater.*, **2020**, *32*, e1902549).

In light of the referee's points, we fabricated a PG with only PVDF film and compared its performance with that of the composite films. We found that the pure PVDF PG generates a voltage of ~ 25 V and a current of $\sim 2 \mu\text{A}$. The PVDF film with the incorporation of the 10 wt. % EDABCO-CuCl₄ generates a voltage of ~ 63 V and current of $\sim 6 \mu\text{A}$. Therefore, compared with a pure PVDF device, the incorporation of the EDABCO-CuCl₄ nanoparticles increased the output voltage and current density by a factor of 2 and 3, respectively. In addition, the measured output power density of the 10 wt. % EDABCO-CuCl₄@PVDF PG is appreciably enhanced compared with that of the pure PVDF PG (43 vs. $1.5 \mu\text{W}/\text{cm}^2$), shown below and Figure S13. Therefore, PVDF and our samples show significantly different contributions to the performance of the piezoelectric nanogenerator.

Supplementary Figure S13. (a) The instantaneous power density of a pure PVDF PG with load resistance ranges from 0 to 50 MΩ. (b) Comparison of the power density of a pure PVDF and EDABCO-CuCl₄@PVDF PG.

2. Did the authors look into the phase of PVDF in their composites? As PVDF is piezoelectric for one of its forms, the authors should specify the actual phase of PVDF in their composites.

Response: We used FTIR transmittance measurement to explore the phase of PVDF in the 10 wt. % EDABCO-CuCl₄@PVDF. The characteristic peaks at 445, 510 and 840 cm^{-1} correspond to the beta-phase PVDF.

We have added the FTIR data in the revised manuscript in Figure S15.

Supplementary Figure S15. FTIR transmission spectra of the 10 wt. % EDABCO-CuCl₄@PVDF film. The characteristic peaks at 445, 510 and 840 cm⁻¹ correspond to the beta-phase of PVDF.

3. Additionally, the authors should highlight the increase in the output performance of PVDF after dispersing their crystallites in PVDF. Because from Figure 3b it is evident that PVDF itself is giving a voltage of 25 V. So, to the reviewer's knowledge, the actual output of the compound can be stated as only 38 V.

Response: In response to this comment, we have added the following sentence to the revised manuscript (page 9):

“Compared with the case of a pure PVDF device, the incorporation of the EDABCO-CuCl₄ nanoparticles increased the output voltage and current density by a factor of 2 and 3, respectively.”

4. Even the power density has almost 40 % contribution from PVDF itself. So it is ambiguous to say that this molecule gives an extraordinary power density of 43 μW/cm². Since there are several molecular nanogenerator devices reported, both with PVDF and other commercial amorphous polymers like poly dimethyl siloxane (PDMS) with some fair power density values, The authors should discuss all these parameters broadly to claim their highest power density.

Response: We measured the output power density of pure PVDF PG, which shows a maximum output power density of 1.5 μW/cm², at 5 Hz and 15 N applied force. At the same external excitation, the measured output power density of the 10 wt. % EDABCO-CuCl₄@PVDF PG is 43 μW/cm², shown below and Figure S13.

We also compared the normalized output power density value of the EDABCO-CuCl₄@PVDF by discarding the maximum power density contributed by the PVDF PG. The normalized power density value of 0.55 remains to be well above the composite PGs reported in Table S2.

We have added the following sentence to the revised manuscript

“The maximum power density of the pure PVDF PG was measured to be 1.5 $\mu\text{W}/\text{cm}^2$ at a threshold load resistance of 2 $\text{M}\Omega$ (Figure S13).” (page 10)

We have added the following notes in the supporting information Table S2

“The measured output power density from the PVDF PG was 1.5 $\mu\text{W}/\text{cm}^2$. By discarding the contribution from pure PVDF, the normalized power density of the EDABCO-CuCl₄-PVDF PG becomes 0.55.”

Supplementary Figure S13. (a) The instantaneous power density of a pure PVDF PG with load resistance ranges from 0 to 50 $\text{M}\Omega$. (b) Comparison of the power density of a pure PVDF and EDABCO-CuCl₄@PVDF PG.

5. Again. The statement “The 10 wt. % piezoelectric generators (which contain only 5 mg of EDABCO-CuCl₄) produce a maximum power density of 43 $\mu\text{W}/\text{cm}^2$ at a threshold load resistance of 8 $\text{M}\Omega$; the best reported for a molecular perovskite energy harvester” by the authors is not justifying the output of the material solely, as PVDF has its own contributions.

Response: In light of this comment, we have further measured the maximum output power density generated from the pure PVDF PG. We have also discussed the contribution of the PVDF in the output power density both in the manuscript and supporting information.

6. How did the authors minimize the static charge developments on the surface of the device during the piezoelectric measurements?

Response: We used a custom-made linear motor with a metallic hammer to apply force on the device, and simultaneously, the body of the hammer was grounded. We attached copper foil tape on top of the polyester packaged device, which will minimize the static charge build-up on the surface. Although metal-to-metal contact can create very few static charges due to the surface curvature, it will be minimized significantly compared to the direct contact between metallic hammer and dielectric-type packaging material (*Mater. Today*, **2019**, *30*, 34-51). We also performed the output polarity switching test to prove that the signal comes from the intrinsic piezoelectric effect of our PGs, thus discarding the electrostatic charging phenomena on the surface of the packaging material.

We have added the following sentence in the experimental section (page15):

“Copper tape was attached to one of the surfaces of the thermal laminating pouches to minimize static charge build up during the periodic application of force by a metallic hammer”

7. The authors state that “With increasing temperature (273 K to 433 K) the V_{OC} of the piezoelectric generators does not decrease, nor does it alter after ~2,500 working cycles, indicating its operability over a large temperature range (Figure 3e)”. For the temperature-dependent Piezoelectric measurements, the authors chose PDMS, why? The authors could have chosen PDMS itself for all their nanogenerator measurements, the use of two different polymers (PVDF and PDMS) should be explained properly.

Response: PDMS is a heat-resistant flexible polymer, and its thermal degradation occurs at a temperature above 400 °C, while PVDF melts at ~ 160 °C. We now better explain that this reasoning lay behind our choice of PDMS for the high-temperature measurement. However, PDMS has some problems related to energy harvesting. Firstly, PDMS has a low elastic modulus (1-3 MPa) which makes it very fragile to sustain high force for energy generation. Secondly, PDMS is very viscous, which will impede uniform nanoparticle distribution. It is difficult to control the nanoparticle dispersion inside the PDMS matrix. Lowering the concentration of PDMS in the solvent will further reduce its elastic modulus. Previous study by Ding *et al.* revealed that output voltage and current density increase by a factor of 3.5 and 2 by changing the host polymer from the PDMS to the PVDF (*Nano Energy*, **2017**, *37*, 126-135; *Adv. Funct. Mater.* **2016**, *26*, 7708-7716). Thirdly, PDMS has very high electrical resistivity ($2.9 \times 10^{12} \Omega \cdot m$) (*Prog. Polym. Sci.*, **2018**, *83*, 97-134), whereas a lower resistance to electrical current in a circuit is desirable for a power source to produce high power. Hence, we did not use PDMS for power source demonstrations.

We have added the following sentences to the revised manuscript:

- “Polymer composites with 5 wt. % EDABCO-CuCl₄ were prepared by dispersing the nanoparticles in a polymer of polydimethylsiloxane (PDMS). PDMS is a heat resistant flexible polymer where thermal degradation occurs at > 400 °C, in contrast to the low melting point of the PVDF (~ 160 °C).” (page 11)
- “PVDF is a stiff polymer with an excellent mechanical strength (elastic modulus 0.03-17.1 GPa), and has low viscosity for proper nanoparticle dispersion.” (page 9)

8. The space group- $P2_12_12_1$ only allows for shear piezoelectricity ($d_{14}/d_{25}/d_{36}$). So, what could be the reason for this high piezoelectric coefficient value i.e 165 pm/V?

Response: In this work, we report the longitudinal piezoelectric coefficient (d_{33}^*), which is equal to $d_{14} + d_{25} + d_{36}$. We speculate that the high piezoelectric coefficient is due to mechanical flexibility, something manifested in Young’s modulus measurements. The Young’s modulus and Hardness of EDABCO-CuCl₄ are measured to be 14.9 ± 0.7 GPa and 580 ± 30 MPa. This modulus value is smaller than those of the piezoelectric ceramics, such as BaTiO₃ (E = 100 ~130 GPa, H = 4~5 GPa) and PZT (E = 60 ~90 GPa, H = 4~7 GPa), indicating a more favourable mechanical flexibility and softness than inorganic ceramics as a lower modulus enables higher displacement at lower stress levels. Therefore, we partially attribute the high piezoelectric coefficient to the increased mechanical flexibility.

We have added the above discussion to the revised manuscript.

9. Jahn-Teller distortion is very well known in many of the organic-inorganic hybrid perovskites, so, Can the authors specify the novelty in this material?

Response: We now discuss more clearly what is new in the present work relative to prior studies. First, we demonstrate that choosing a quasi-spherical A cation and Jahn-Teller active B cation significantly contributes to the chemical design as well as the performance of molecular piezoelectric. Second, this material is an environmentally friendly molecular piezoelectric with a large piezoelectric coefficient and piezoelectric voltage coefficient. Third, the piezoelectric effect and the corresponding piezoelectric nanogenerator response show high thermal stability. In particular, we measure stability above 150°C, which is the standard maximum temperature considered in automotive applications. This work provides a guide

to assist the discovery of new molecular piezoelectrics, enabling targeted design rather than a blind search, and contributes to the discovery of high-performance piezoelectric energy harvesters.

10. What is the melting point of the compound?

Response: On page 7 of the revised manuscript, we now provide that the melting process of EDABCO-CuCl₄ occurs at 200 °C (inset of Figure S5a).

11. It is mentioned that there is no phase transition in this molecule, but the variable temperature PXRD data clearly shows that some changes are happening in the molecule before 200 °C. The authors should describe the changes in the PXRD pattern, where new peaks are appearing after a certain temperature (117 to 121 line)

Response: In light of the referee's points, we now discuss the changes in the PXRD patterns.

We used TG-DSC measurement to characterize the phase transition in EDABCO-CuCl₄ (Figure S5a). The DSC curve shows endothermic/exothermic peaks at 200-226 °C, due to the melting and the subsequent decomposition of EDABCO-CuCl₄. The decomposition process is seen in weight loss at 210 °C in the TG curve. We attribute the changes in the PXRD patterns at 200 °C to the decomposition of EDABCO-CuCl₄, where a 10 °C temperature difference is due to the difference in the equipment and environment.

We have added the above discussion to the revised manuscript (page 7).

12. The authors mentioned that “With increasing temperature (273 K to 433 K) the V_{OC} of the piezoelectric generators does not decrease, nor does it alter after ~2,500 working cycles, indicating its operability over a large temperature range (Figure 3e).” Figure 3e shows that the cyclic stability measurements at 433K result in a change in the voltage (~20 V to 30 V), what could be the reason for this? And why the authors are still claiming that there is no change associated.

Response: We now provided the zoom-in figure of the V_{oc} at 433 K, which shows the average magnitude of the maximum output voltage is still ~ 20 V (please see the figure below). We anticipate that the adhesive's bonding strength was declining due to its prolonged exposure to elevated temperatures while simultaneously absorbing the mechanical force. Therefore, a slight but periodic displacement of the device

under the applied force created spikes in the output. The conductivity of the electrodes changes at elevated temperatures and could create noise peaks in the output.

We have clarified the change in the output at high temperatures by quantifying the deviation in the revised manuscript:

“With increasing temperature (273 K to 433 K) the average V_{oc} of the piezoelectric generators remains stable (~ 20 V) after $\sim 2,500$ working cycles, indicating its operability over a large temperature range (Figure 3e). The device and material exhibited no noticeable degradation following high-temperature operation (cross-sectional SEM and EDS mapping in Figure S14). We noticed some noise-like spikes in the output signal that could be due to the declining adhesive strength that was used to mount the device on the electric heater.” (Page 11)

Figure R1. Reliability test of EDABCO-CuCl₄@PDMS composite films for $\sim 2,500$ cycles at 433 K.

13. Please provide the method employed for the construction of the surface and provided dipole moment values in Figure 1a.

Response: The electrostatic potential and dipole moments of the ligands were calculated using the Gaussian 09 package at the B3LYP/def2TZVP level with DFT-D3. The electrostatic potential plot was visualized in GaussianView. As shown in Figure 1a, the calculated dipole moment of DABCO and EDABCO are 0 and 3.64 Debye, respectively.

We have added the above calculation details in the Methods.

Main manuscript Figure 1a. The modification of a spherical cation into a quasi-spherical cation lowers molecular symmetry and enables dipole moment.

Reviewer #3 (Remarks to the Author):

Piezoelectrics with both large d_{33} and large g_{33} are highly desirable for piezoelectric energy harvesting. In this manuscript, the authors reported a non-ferroelectric organic-inorganic hybrid Cu(II) halide piezoelectric EDABCO-CuCl₄, which shows a d_{33} and g_{33} up to 165 pm/V and $\sim 2650 \times 10^{-3}$ V m N⁻¹, respectively. The EDABCO-CuCl₄-based piezoelectric energy harvester exhibits excellent performance with a high peak power density of 43 μ W/cm². It is known that large d_{33} piezoelectrics are generally based on ferroelectrics. The discovery of a non-ferroelectric with such a high d_{33} of 165 pm/V is interesting. I think this work gives new insights into the piezoelectric field and provides a good strategy for the exploration of high-performance piezoelectrics. However, before publication in Nature Communications, the following issues should be addressed.

1. Although there are many kinds of perovskite structures ranging from 0D-3D, they are derived from the typical perovskite CaTiO₃ and basically have octahedral configurations in the structure. Therefore, it is hard to regard EDABCO-CuCl₄ with [CuCl₄]²⁻ tetrahedrons as perovskites.

Response: We have changed “perovskite” to “metal halide” in the revised manuscript.

2. The authors also mentioned the enhanced piezoelectric response in the title. But the DABCO-CuCl₄ and EDABCO-ZnCl₄ used for comparison have no piezoelectric response.

Response: We used DABCO-CuCl₄ and EDABCO-ZnCl₄ to demonstrate that both the quasi-spherical cation and the Jahn-Teller distortion are connected with the non-centrosymmetric structure.

To make the expression more precise, we now changed the title to “Large piezoelectric response in a Jahn-Teller distorted molecular metal halide”

3. In the abstract, the authors stated that the quasi-spherical cation into a Jahn-Teller distorted perovskite lattice increases the mechanical strain for a large piezoelectric response, while the strain data was not provided.

Response: We have now provided Young’s modulus and Hardness data that seek to investigate the impact of the combining quasi-spherical cation and Jahn-Teller distortion on mechanical response, shown below and Figure S7. The Young’s modulus and Hardness of EDABCO-CuCl₄ have been measured as 14.92 ± 0.7 GPa and 581 ± 30 MPa. This modulus value is smaller than those of the piezoelectric ceramics, such as BaTiO₃ (E = 100 ~130 GPa, H = 4~5 GPa) and PZT (E = 60 ~90 GPa, H = 4~7 GPa), indicating more favourable mechanical flexibility and softness than inorganic ceramics. Meanwhile, the modulus of EDABCO-CuCl₄ is higher than DABCO-CuCl₄ and EDABCO-ZnCl₄, which allows it to be robust for repeated cycling performance.

The full revised context is shown below and in the Main text:

- “With this in mind, we sought to insert a quasi-spherical cation into a Jahn-Teller distorted lattice, increasing mechanical response for a large piezoelectric coefficient.” (page 2)
- “We attribute the high piezoelectric coefficient of EDABCO-CuCl₄ to the increased mechanical flexibility, as manifested in Young’s modulus measurements (Figure S7). The Young’s modulus of EDABCO-CuCl₄ is significantly smaller than those of typical piezoelectric ceramics, indicating favorable mechanical flexibility, but higher than other non-piezoelectric metal halides, which allows it to be robust for repeated cycling performance.” (page 8)

Supplementary Figure S7. Young's modulus and hardness of DABCO-CuCl₄, EDABCO-CuCl₄, and EDABCO-ZnCl₄ single crystals.

4. The strategy by modification of a spherical cation into a quasi-spherical cation to lower molecular symmetry to regulate the crystal symmetry has recently been proposed as the quasi-spherical theory (*J. Am. Chem. Soc.* **2020**, *142*, 15205-15218). This should be mentioned in the manuscript.

Response: We have improved the discussion on page 3 of quasi-spherical theory:

“We see that quasi-spherical theory, which explains how subtle modifications to spherical molecules can impart symmetry breaking into this material system and cause significant polarization changes under external stress, can thereby give rise to large d_{33} .” (Page 4)

We have added the recommended paper as Ref. 14 in the revised manuscript.

5. The authors obtained the d_{33} along the [111] direction. Please provide the corresponding data to verify the [111] direction.

Response: We used XRD to determine the direction of the crystal surface, shown below and in Figure S6. The largest natural face is (001) direction, and the (111) direction is cut along the ab -plane with a 45° derivation to the c direction.

Supplementary Figure S6. (c) Bulk single crystal of EDABCO-CuCl₄, the largest natural face is the (001) direction. (d) Schematic of the (111) direction in the crystal. The morphology is calculated according to the single-crystal structure data.

6. Generally, the calculation of g_{33} is based on the ϵ_r at the frequency of 1 kHz.

Response: In light of the referee's feedback, the g_{33} at 1 kHz is calculated as $2110 \times 10^{-3} \text{ V m N}^{-1}$ with an ϵ_r of 8.83. We have updated the g_{33} and $d_{33} \cdot g_{33}$ values in the revised manuscript on page 9 and Table S1.

7. The authors prepared the EDABCO-CuCl₄@PVDF composite film-based piezoelectric generator. However, PVDF is also piezoelectric. How to rule out the influence of PVDF.

Response: To address this comment, we fabricated a PG with the PVDF only, and compared its performance with the composite films, shown below and Figure S13. We found that the pure PVDF PG generates a maximum output voltage of $\sim 25 \text{ V}$ and a current of $\sim 2 \mu\text{A}$. The PVDF film with the incorporation of the 10 wt. % EDABCO-CuCl₄ generates a voltage of $\sim 63 \text{ V}$ and current of $\sim 6 \mu\text{A}$. In response to the reviewer's suggestion, we also measured the output power density of the pure PVDF PG. We found that the PVDF PG produces a maximum output power density of $1.5 \mu\text{W}/\text{cm}^2$, considerably lower than the 10 wt. % EDABCO-CuCl₄ PG ($43 \mu\text{W}/\text{cm}^2$). Therefore, PVDF and our samples show significantly different contributions to the performance of the piezoelectric nanogenerator.

Supplementary Figure S13. (a) The instantaneous power density of a pure PVDF PG with load resistance ranges from 0 to 50 $\text{M}\Omega$. (b) Comparison of the power density of a pure PVDF and EDABCO-CuCl₄@PVDF PG.

8. In the experimental section, for the preparation of EDABCO-CuCl₄ single crystals, 1-Ethyl-1,4-diazabicyclo [2.2.2] octanium iodide was used. The authors are suggested to check whether the iodide ion was doped into the single crystals. It is better to use the 1-Ethyl-1,4-diazabicyclo [2.2.2] octanium chloride to prepare EDABCO-CuCl₄.

Response: We used X-ray photoelectron spectroscopy (XPS) measurement to check whether the iodide ion was doped into EDABCO-CuCl₄ single crystals (Figure R2). The XPS spectra show that there is no peak corresponding to the I 3*d* orbital in the binding energy between 640 and 610 eV, whereas distinct peaks of Cl 2*p* orbitals are observed. This result indicates that there is no iodide ion in the EDABCO-CuCl₄ single crystal. In the single crystal process, we use hydrochloric acid as the solvent, and the content of Cl is excessive.

Figure R2. High-resolution XPS spectra of (a) I 3*d* and (b) Cl 2*p*.

9. The authors used high-voltage electrical poling to align the dipoles in the EDABCO-CuCl₄@PVDF film. But EDABCO-CuCl₄ is a non-ferroelectric.

Response: Previously, we used electrical poling as a general protocol for the PG fabrication, we now suggest that the poling is not necessary as EDABCO-CuCl₄ is a non-ferroelectric material. We measured the output voltage of the EDABCO-CuCl₄@PVDF PG before and after applying the electrical poling field of 50 V/μm for 2-3 hours. We found that the output voltage does not change with the poling field (Please see the figure below and Figure S17).

We have added the following sentence on page 14 of the revised manuscript:

“To explore the influence of poling on the output performance, we measured the output voltage of the EDABCO-CuCl₄@PVDF film before and after applying an electrical poling field of 50 V·μm⁻¹ for 2–3 hours, and the results showed that poling did not affect the output performance (Figure S17).”

Supplementary Figure S17. Comparison of the output voltage of the unpoled and poled EDABCO-CuCl₄@PVDF film.

10. In Table S1, some molecular piezoelectrics with large d_{33} and large g_{33} were not listed, such as the C₆H₅N(CH₃)₃CdBr₂Cl_{0.75}I_{0.25} (*Nat. Commun.* **2022**, *13*, 5607).

Response: We have added C₆H₅N(CH₃)₃CdBr₂Cl_{0.75}I_{0.25} and the corresponding data to Table S1 in the revised manuscript.

REVIEWER COMMENTS

Reviewer #1 (Remarks to the Author):

The authors have adequately addressed my comments.

Reviewer #2 (Remarks to the Author):

The authors have revised the manuscript with new experiments and clear explanations. They have mentioned that they chose to make their material with PVDF composites due to their temperature compatibility and the problems associated with making composites with certain other polymers such as PDMS. To demonstrate the role of their composite for the output performance, they have shown the activity of the neat PVDF device and compared the output. However, this method is not accurate, especially for PVDF as external additives are known to enhance the content of the piezoelectrically active beta-phase of PVDF. The authors have cited some previous references, where there were reports of hybrid piezoelectric materials imposed as composites of PVDF. These studies were undertaken at very early stages during the development of this field. Considering numerous publications only with PVDF and the inherent issues of using PVDF for determining the piezoelectric nature of other materials, I recommended using other polymers. Hence, it is better to show a model device with another polymer (say PVA, PCL, etc.) and show that the activity of that new polymer composite is comparable to that of their PVDF polymer composite. Some more additional points the authors must address before the manuscript can be suitable for publication in Nature Communications.

- 1) In the absence of a comparison with other polymer composites, please provide some theoretical studies to show the difference in the piezoelectric activity of their compound and PVDF and distinguish the contribution of your compound from that of PVDF for the observed high Voc.
- 2) With increasing temperature there is clearly a change in Voc of about 5-10 V. This anomaly is not explained in the answers. The change in Voc should be addressed properly.
- 3) To comment on the beta phase of PVDF, FTIR experiments have been performed on the composite film. To distinguish between the polymer and the compound, the authors must perform the same on their pure samples and demonstrate the presence of the compound and PVDF on the film.

Reviewer #3 (Remarks to the Author):

In this version, the authors have made comprehensive revisions to address reviewers' concerns. The manuscript has been much improved. The discovery of a non-ferroelectric with a high d_{33} is interesting. I would be pleased to recommend the acceptance of the manuscript in the current version.

Point-by-point response

Reviewer #2 (Remarks to the Author):

The authors have revised the manuscript with new experiments and clear explanations. They have mentioned that they chose to make their material with PVDF composites due to their temperature compatibility and the problems associated with making composites with certain other polymers such as PDMS. To demonstrate the role of their composite for the output performance, they have shown the activity of the neat PVDF device and compared the output. However, this method is not accurate, especially for PVDF as external additives are known to enhance the content of the piezoelectrically active beta-phase of PVDF. The authors have cited some previous references, where there were reports of hybrid piezoelectric materials imposed as composites of PVDF. These studies were undertaken at very early stages during the development of this field. Considering numerous publications only with PVDF and the inherent issues of using PVDF for determining the piezoelectric nature of other materials, I recommended using other polymers. Hence, it is better to show a model device with another polymer (say PVA, PCL, etc.) and show that the activity of that new polymer composite is comparable to that of their PVDF polymer composite.

Some more additional points the authors must address before the manuscript can be suitable for publication in Nature Communications.

1) In the absence of a comparison with other polymer composites, please provide some theoretical studies to show the difference in the piezoelectric activity of their compound and PVDF and distinguish the contribution of your compound from that of PVDF for the observed high V_{oc} .

Author Actions in Response: In light of the referee's request for theoretical studies of the different in piezoelectric activity, we now provide a finite-element simulation of the projected piezoelectric potential.

The results of the new simulation are included as Figure S18 in the Supplementary Information. We also include the same new figure below.

For the pure PVDF film, the calculated piezoelectric potential is 0.04 V at an applied pressure of 50 kPa. After introducing nanoparticles (approximately 10% of PVDF) to form the composite film, the calculated piezoelectric potential changes to 0.1 V, 2.5 times that of the pure PVDF film. This value is consistent with the value seen in the 10% EDABCO-CuCl₄@PVDF device, where a ratio of ~2.5 is obtained by comparing the V_{oc} of the composite film and pure PVDF.

Figure R1: Piezoelectric potential distribution in (a) pure PVDF and (b) 10% EDABCO-CuCl₄@PVDF composite film.

2) With increasing temperature there is clearly a change in V_{oc} of about 5-10 V. This anomaly is not explained in the answers. The change in V_{oc} should be addressed properly.

Author Actions in Response: We have added the following discussion to page 11 of the revised manuscript:

“We sought to test the applicability of EDABCO-CuCl₄ for use in sensing and actuating in harsh environments. Polymer composites with 5 wt. % EDABCO-CuCl₄ were prepared by dispersing the nanoparticles in polydimethylsiloxane (PDMS). PDMS is a heat-resistant flexible polymer for which thermal degradation occurs above 400 °C, in contrast to lower-melting-point PVDF (~ 160 °C). The average value of V_{oc} of ~ 20 V is maintained following 2,500 working cycles at 433 K, indicating its operability over a widened temperature range (Figure 3e). When we increase temperature from 273 K to 433 K, we see an increase in the V_{oc} (± 5 V). This increase, as well as higher pickup of ambient noise observed in Figure 3e, warrant further mechanistic study, including with a focus of the impact of temperature on both electrical conductivity and adhesive strength. The device and material exhibited no noticeable degradation following high-temperature operation (cross-sectional SEM and EDS mapping in Figure S14).”

3) To comment on the beta phase of PVDF, FTIR experiments have been performed on the composite film. To distinguish between the polymer and the compound, the authors must perform the same on their pure samples and demonstrate the presence of the compound and PVDF on the film.

Author Actions in Response: In the revised supplementary information (new Fig. S15), we now provide FTIR transmission spectra of EDABCO-CuCl₄ and PVDF, in addition to FTIR spectrum previously presented for EDABCO-CuCl₄@PVDF film. The new data show that the FTIR peaks of EDABCO-CuCl₄ and PVDF are seen in the EDABCO-CuCl₄@PVDF film. We now report the estimated β -phase content for pure PVDF (80%) and for the composite film (85%). Given that pure PVDF generates a V_{oc} of 25 V, we associate the contribution from the increased 5% β -phase PVDF to be roughly 1.5 V, whereas an increase of 38 V is observed for the composite film. We now comment in the revised work that changes in the β -phase PVDF account for < 5% of the enhancement of the V_{oc} in the composite film.

REVIEWERS' COMMENTS

Reviewer #2 (Remarks to the Author):

I reviewed the revised manuscript. The authors have responded only to the additional points I asked them to address and not to the main point given at the end of the comments paragraph that reads below.

"Considering numerous publications only with PVDF and the inherent issues of using PVDF for determining the piezoelectric nature of other materials, I recommended using other polymers. Hence, it is better to show a model device with another polymer (say PVA, PCL, etc.) and show that the activity of that new polymer composite is comparable to that of their PVDF polymer composite.

Please respond to this comment either with additional experiments or with a justification of why no other polymers such as PVA, PCL or TPU are not suitable.

R#1's comment on the report of R#2

Reviewer 2's comments are helpful to improve this work. The authors have provided detailed response and revision to the comments. Thus, there is no need to make additional composite samples for this study.

It is suggested that the authors could add additional discussion to compare the performance between PVDF-based and PDMS-based devices, which are already studied in this work.

The discussion would be also helpful to address the concern from Reviewer 2.

R#3's comment on the report of R#2

I am glad to review the article entitled "Large piezoelectric response in a Jahn-Teller distorted molecular metal halide". I have carefully read this article again and the comments of Reviewer 2. The main point of this work is the discovery of a Jahn-Teller distorted organic-inorganic hybrid metal halide piezoelectric EDABCO-CuCl₄ with both outstanding d₃₃ and g₃₃, which I think has sufficient significance and novelty for publication in Nature Communications. The authors also prepared the EDABCO-CuCl₄@PVDF composite film-based piezoelectric generator, which shows excellent energy harvesting performances with a high output power density. However, PVDF is a piezoelectric polymer. It is hard to clearly distinguish between the contributions of PVDF and EDABCO-CuCl₄, as pointed out by Reviewer 2. Reviewer 2 then recommended that it is better to show a model device by using other non-piezoelectric polymers such as PVA, PCL or TPU, and show that the activity of that new polymer composite is comparable to that of the PVDF polymer composite. In my opinion, if possible, I also suggest the authors try to prepare a piezoelectric generator device with a non-piezoelectric polymer to better show the energy harvesting performance of EDABCO-CuCl₄. However, it is not necessary to obtain comparable energy harvesting performances to the EDABCO-CuCl₄@PVDF composite device since the performance is affected by many factors such as processing technique, device structure, interface, and the homogeneity of the composite, which needs a long-term exploration and optimization to obtain a high value. At least, the authors are recommended to add some discussions about why other non-piezoelectric polymers such as PVA, PCL or TPU are not suitable, as also suggested by Reviewer 2. In addition, I also suggest the authors make their statement clear that the high output power density is based on the EDABCO-CuCl₄@PVDF composite in the abstract and conclusion parts.

Point-by-point response

Reviewer #2 (Remarks to the Author):

I reviewed the revised manuscript. The authors have responded only to the additional points I asked them to address and not to the main point given at the end of the comments paragraph that reads below.

"Considering numerous publications only with PVDF and the inherent issues of using PVDF for determining the piezoelectric nature of other materials, I recommended using other polymers. Hence, it is better to show a model device with another polymer (say PVA, PCL, etc.) and show that the activity of that new polymer composite is comparable to that of their PVDF polymer composite.

Please respond to this comment either with additional experiments or with a justification of why no other polymers such as PVA, PCL or TPU are not suitable.

Reply: In response to this comment, we have added the following discussion to the manuscript (page 15):

“The Young’s modulus disparity between the host matrix and the NPs should be minimized in order to maximize the efficiency of stress transfer to the NPs. PVDF has Young’s modulus value (0.03-17.1 GPa) that is much closer to that of EDABCO-CuCl₄ (14.7± 0.7 GPa) compared to the values in other polymers such as polyvinyl acetate (PVA), polycaprolactone (PCL), and thermoplastic polyurethane (TPU). We selected PVDF to reinforce the EDABCO-CuCl₄ NPs with this in mind.”

Here we elaborate further on this topic:

In piezoelectric composites, energy harvesting performance is determined by the efficiency of load transfer from the surrounding polymer matrix to the active piezoelectric filler. When an external mechanical stimulus is applied, the nanoparticles act as stress-concentrating centers and generate piezoelectric polarization. Thus, it is crucial to ensure efficient stress transfer to the nanoparticles, which is impacted by the disparity in Young's modulus (YM) between the host matrix and the nanoparticles (NPs) (*Energy Environ. Sci.*, 2018, **11**, 2046):

$$\eta_{NP} = \frac{k}{1 - \phi^3}$$

Here, k represents the ratio of YM of the host matrix to that of the NPs, while ϕ is the volume fraction of NPs.

The load transfer efficiency of different host matrices is calculated as follows:

Case 1: PVA as a host matrix

Polyvinyl acetate (PVA) has a YM of 380 MPa (*Chem. Mater.* **2003**, *15* (26), 5019–5024), whereas the measured YM of EDABCO-CuCl₄ is 14.7 ± 0.7 GPa. Therefore, the calculated k value is ~ 0.025, which is quite low. In addition, the presence of hydroxyl groups in PVA causes it to absorb moisture from the ambient, leading to mechanical reliability issues due to film swelling.

Case 2: PCL as a host matrix

Polycaprolactone (PCL) is a biodegradable polymer with a YM of 364.3 MPa, and the calculated k value is ~ 0.024. However, PCL has a low melting point of 60 °C, which limits its application (*Acta Biomaterialia*, **2010**, *6* (7), 2467-2476).

Case 3: TPU as a host matrix

Thermoplastic polyurethane (TPU) has a YM of 53.3 MPa (*J. Mater. Chem.*, **2012**, *22*, 11748). The k value, therefore, scales to ~ 0.003.

Case 4: PVDF as a host matrix

The YM of PVDF varies between 0.03 and 17.1 GPa and is measured as 0.9 GPa (900 MPa) in our lab (Figure S11b). The k value is 0.061 when using PVDF as the host matrix, which is much higher than that in PVA, PCL, and TPU polymers.

It is important to note that Young's modulus disparity (k) will have a significant impact on the polarization of EDABCO-CuCl₄ nanoparticles in response to applied pressure. Therefore, when designing the host matrix for a piezoelectric nanoparticle composite, it is crucial to take into account the disparity of Young's modulus.

R#1's comment on the report of R#2

Reviewer 2's comments are helpful to improve this work. The authors have provided detailed responses and revision to the comments. Thus, there is no need to make additional composite samples for this study.

It is suggested that the authors could add additional discussion to compare the performance between PVDF-based and PDMS-based devices, which are already studied in this work.

The discussion would be also helpful to address the concern from Reviewer 2.

Reply: We thank the reviewer for this suggestion. In response to this comment, we have added the following sentences to the supporting information (page 16):

“Compared to the output voltage of ~ 63 V in the 10 wt. % composite with PVDF, the 5 wt. % EDABCO-CuCl₄@PDMS generates an output voltage of ~ 20 V. This is a quantity that could potentially be enhanced by increasing the nanoparticle concentration and controlling their dispersion. We also note that the lower Young’s modulus of PDMS (1.4 MPa) than PVDF will consume more mechanical stress and reduce the stress transfer efficiency to the EDABCO-CuCl₄ NPs.”

R#3’s comment on the report of R#2

I am glad to review the article entitled “Large piezoelectric response in a Jahn-Teller distorted molecular metal halide”. I have carefully read this article again and the comments of Reviewer 2. The main point of this work is the discovery of a Jahn-Teller distorted organic-inorganic hybrid metal halide piezoelectric EDABCO-CuCl₄ with both outstanding d_{33} and g_{33} , which I think has sufficient significance and novelty for publication in Nature Communications. The authors also prepared the EDABCO-CuCl₄@PVDF composite film-based piezoelectric generator, which shows excellent energy harvesting performances with a high output power density. However, PVDF is a piezoelectric polymer. It is hard to clearly distinguish between the contributions of PVDF and EDABCO-CuCl₄, as pointed out by Reviewer 2. Reviewer 2 then recommended that it is better to show a model device by using other non-piezoelectric polymers such as PVA, PCL or TPU, and show that the activity of that new polymer composite is comparable to that of the PVDF polymer composite. In my opinion, if possible, I also suggest the authors try to prepare a piezoelectric generator device with a non-piezoelectric polymer to better show the energy harvesting performance of EDABCO-CuCl₄. However, it is not necessary to obtain comparable energy harvesting performances to the EDABCO-CuCl₄@PVDF composite device since the performance is affected by many factors such as processing technique, device structure, interface, and the homogeneity of the composite, which needs a long-term exploration and optimization to obtain a

high value. At least, the authors are recommended to add some discussions about why other non-piezoelectric polymers such as PVA, PCL or TPU are not suitable, as also suggested by Reviewer 2. In addition, I also suggest the authors make their statement clear that the high output power density is based on the EDABCO-CuCl₄@PVDF composite in the abstract and conclusion parts.

Reply: We have added discussion to the main manuscript and supporting information. We have also revised our statement in the abstract and conclusion sections.

Action 1: We added the following discussion in the main manuscript (page 15)

“The Young’s modulus disparity between the host matrix and the NPs should be minimized in order to maximize the efficiency of stress transfer to the NPs. PVDF has Young’s modulus value (0.03-17.1 GPa) that is much closer to that of EDABCO-CuCl₄ (14.7± 0.7 GPa) compared to the values in other polymers such as polyvinyl acetate (PVA), polycaprolactone (PCL), and thermoplastic polyurethane (TPU). We selected PVDF to reinforce the EDABCO-CuCl₄ NPs with this in mind.”

Action 2: We added the following discussion in the supporting information (page 16) which was suggested by Reviewer #1

“Compared to the output voltage of ~ 63 V in the 10 wt. % composite with PVDF, the 5 wt. % EDABCO-CuCl₄@PDMS generates an output voltage of ~ 20 V. This is a quantity that could potentially be enhanced by increasing the nanoparticle concentration and controlling their dispersion. We also note that the lower Young’s modulus of PDMS (1.4 MPa) than PVDF will consume more mechanical stress and reduce the stress transfer efficiency to the EDABCO-CuCl₄ NPs.”

Action 3: We have changed our statement in the abstract and conclusion:

“This enables piezoelectric energy harvesting in EDABCO-CuCl₄@PVDF (polyvinylidene fluoride) composite film with a peak power density of 43 μW/cm² (at 50 kPa), the highest value reported for mechanical energy harvesters based on heavy-metal-free molecular piezoelectric.”
(page 2)

“The 10 wt. % EDABCO-CuCl₄@PVDF composites achieve output power densities superior to the best-reported molecular hybrid energy harvester.” (page 11)

In addition, we have distinguished the individual contribution of PVDF and EDABCO-CuCl₄ in energy generation by the following experiments:

- (i) We have measured the β -phase percentage in pure PVDF and the composite film to identify whether there is any contribution of the NPs to improve the β -phase of PVDF.
- (ii) We have fabricated PGs with pure PVDF to compare the output voltage, current, and power density before and after the nanoparticle incorporation.

We hope that our revised manuscript will now clarify all the confusion regarding the role of different polymers in the electromechanical energy conversion process by the PGs. We really appreciate the valuable time, and suggestions provided by our reviewers to improve the quality of this work.